# W2S: Microscopy Data with Joint Denoising and Super-Resolution for Widefield to SIM Mapping

**Abstract.** In fluorescence microscopy live-cell imaging, there is a critical trade-off between the signal-to-noise ratio and spatial resolution on one side, and the integrity of the biological sample on the other side. To obtain clean high-resolution images, one can either use microscopy techniques such as structured-illumination microscopy (SIM), or apply denoising and super-resolution (SR) algorithms. However, the former option requires multiple shots that can damage the samples, and although efficient deep learning based algorithms exist for the latter option, no benchmark exists to evaluate these algorithms on the joint denoising and SR (JDSR) tasks.

To study joint denoising and SR on microscopy data, we propose such a novel JDSR dataset, **W**idefield**2S**IM (W2S), acquired using a conventional fluorescence widefield and SIM imaging. W2S includes 144,000 real fluorescence microscopy images, resulting in a total of 360 sets of images. A set is comprised of noisy LR widefield images with different noise levels, a noise-free LR image, and a corresponding high-quality HR SIM image. W2S allows us to benchmark the combinations of 6 denoising methods and 6 SR methods. We show that state-of-the-art SR networks perform very poorly on noisy inputs. Our evaluation also reveals that applying the best denoiser in terms of reconstruction error followed by the best SR method does not necessarily yield the best final result. Both quantitative and qualitative results show that SR networks are sensitive to noise and the sequential application of denoising and SR algorithms is sub-optimal. Lastly, we demonstrate that SR networks retrained end-to-end for JDSR outperform any combination of state-of-the-art deep denoising and SR networks[1].

**Keywords:** Image Restoration Dataset, Denoising, Super-resolution, Microscopy Imaging, Joint Optimization

## 1 Introduction

Fluorescence microscopy allows to visualize sub-cellular structures and protein-protein interaction at the molecular scale. However, due to the weak signals and diffraction limit, fluorescence microscopy images suffer from high noise and limited resolution. One way to obtain high-quality, high-resolution (HR) microscopy images is to leverage super-resolution fluorescence microscopy, such as

---

[1] Code and data available at https://github.com/widefield2sim/w2s

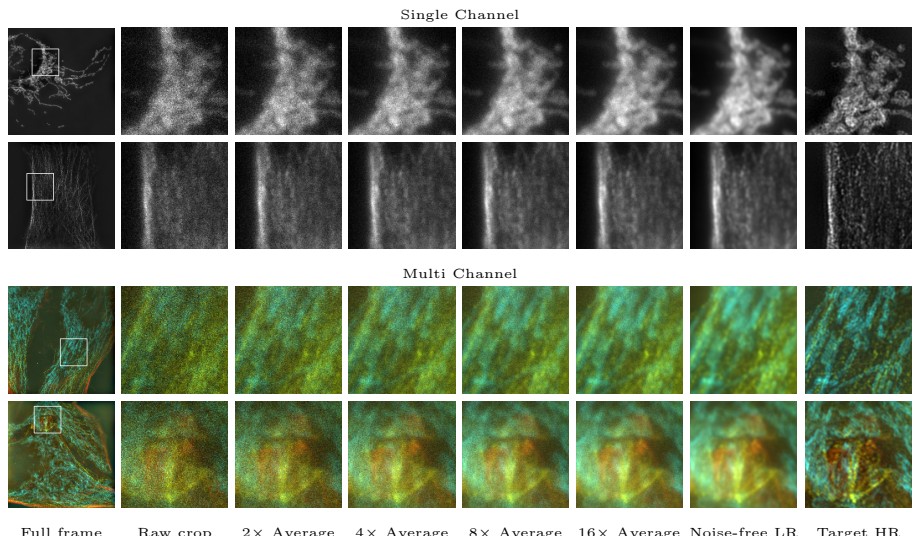

| Full frame | Raw crop | 2× Average | 4× Average | 8× Average | 16× Average | Noise-free LR | Target HR |

**Fig. 1.** Example of image sets in the proposed W2S. We obtain 5 LR images with different noise levels by either taking a single raw image or averaging different numbers of raw images of the same field of view. The more images we average (*e.g.*, 2, 4, 8, and 16), the lower the noise level, as shown in the different columns of the figure. The noise-free LR images are the average of 400 raw images, and the HR images are obtained using structured-illumination microscopy (SIM) [15]. The multi-channel images are formed by mapping the three single-channel images of different wavelengths to RGB. A gamma correction is applied for better visualization. Best viewed on screen.

structure illumination microscopy (SIM) [15]. This technique requires multiple captures with several parameters requiring expert tuning to get high-quality images. Multiple or high-intensity-light acquisitions can cause photo-bleach and even damage the samples. The imaged cells could be affected and, if imaged in sequence for live tracking, possibly killed. This is because a single SIM acquisition already requires a set of captures with varying structured illumination. Hence, a large set of SIM captures would add up to high illumination and an overhead in capture time that is detrimental to imaging and tracking of live cells. Therefore, developing an algorithm to effectively denoise and super-resolve a fluorescence microscopy image is of great importance to biomedical research. However, a high-quality dataset is needed to benchmark and evaluate joint denoising and super-resolution (JDSR) on microscopy data.

Deep-learning-based methods in denoising [2,38,45,11] and SR [42,50,51] today are outperforming classical signal processing approaches. A major limitation in the literature is, however, the fact that these two restoration tasks are addressed separately. This is in great part due to a missing dataset that would allow both to train and to evaluate JDSR. Such a dataset must contain aligned pairs of LR and HR images, with noise and noise-free LR images, to allow retraining

retrain prior denoising and SR methods for benchmarking the consecutive application of a denoiser and an SR network as well as candidate one-shot JDSR methods.

In this paper, we present such a dataset, which, to the best of our knowledge, is the first JDSR dataset. This dataset allows us to evaluate the existing denoising and SR algorithms on microscopy data. We leverage widefield microscopy and SIM techniques to acquire data fulfilling the described requirements above. Our noisy LR images are captured using widefield imaging of human cells. We capture a total of 400 replica raw images per field of view, 8 times more than a recent denoising-only dataset using similar imaging technology [49]. We average several of the LR images to obtain images with different noise levels, and all of the 400 replicas to obtain the noise-free LR image. Using SIM imaging [15], we obtain the corresponding high-quality HR images. Our resulting **W**idefield**2S**IM (W2S) dataset consists of 360 sets of LR and HR image pairs, with different fields of view and acquisition wavelengths. Visual examples of the images in W2S are shown in Fig. 1.

We leverage our JDSR dataset to benchmark different approaches for denoising and SR restoration on microscopy images. We compare the sequential use of different denoisers and SR methods, of directly using an SR method on a noisy LR image, and of using SR methods on the noise-free LR images of our dataset for reference. We additionally evaluate the performance of retraining SR networks on our JDSR dataset. Results show a significant drop in the performance of SR networks when the low-resolution (LR) input is noisy compared to it being noise-free. We also find that the consecutive application of denoising and SR achieves better results. It is, however, not as performing in terms of RMSE and perceptual texture reconstruction as training a single model on the JDSR task, due to the accumulation of error. The best results are thus obtained by training a single network for the joint optimization of denoising and SR.

In summary, we create a microscopy JDSR dataset, W2S, containing noisy images with 5 noise levels, noise-free LR images, and the corresponding high-quality HR images. We analyze our dataset by comparing the noise magnitude and the blur kernel of our images to those of existing denoising and SR datasets. We benchmark state-of-the-art denoising and SR algorithms on W2S, by evaluating different settings and on different noise levels. Results show the networks can benefit from joint optimization.

## 2    Related Work

### 2.1    Biomedical Imaging Techniques for Denoising and Super-resolution

Image averaging of multiple shots is one of the most employed methods to obtain a clean microscopy image. This is due to its reliability and to avoid the potential blurring or over-smoothing effects of denoisers. For microscopy experiments requiring long observation and minimal degradation of specimens, low-light conditions and short exposure times are, however, preferred as multiple shots might

damage the samples. To then reduce the noise influence and increase the resolution, denoising methods and SR imaging techniques are leveraged.

To recover a clean image from a single shot, different denoising methods have been designed, including the Poisson-oriented PURE-LET [25], EPLL [55], and BM3D [7]. Although these methods provide promising results, recent deep learning methods outperform them by a big margin according to our benchmark results. To achieve resolution higher than that imposed by the diffraction limit, a variety of SR microscopy techniques exist, which achieve SR either by spatially modulating the fluorescence emission using patterned illumination (*e.g.*, STED [17] and SIM [15]), or by stochastically switching on and off individual molecules using photo-switchable probes (*e.g.*, STORM [33]), or photoconvertible fluorescent proteins (*e.g.*, PALM [36]). However, all of these methods require multiple shots over a period of time, which is not suitable for live cells because of the motion and potential damage to the cell. Thus, in this work, we aim to develop a deep learning method to reconstruct HR images from a single microscopy capture.

## 2.2  Datasets for Denoising and Super-resolution

Several datasets have commonly been used in benchmarking SR and denoising, including Set5 [3], Set14 [44], BSD300 [27], Urban100 [18], Manga109 [28], and DIV2K [39]. None of these datasets are optimized for microscopy and they only allow for synthetic evaluation. Specifically, the noisy inputs are generated by adding Gaussian noise for testing denoising algorithms, and the LR images are generated by downsampling the blurred HR images for testing SR methods. These degradation models deviate from the degradations encountered in real image capture [5]. To better take into account realistic imaging characteristics and thus evaluate denoising and SR methods in real scenarios, real-world denoising and SR datasets have recently been proposed. Here we discuss these real datasets and compare them to our proposed W2S.

**Real Denoising Dataset**  Only a few datasets allow to quantitatively evaluate denoising algorithms on real images, such as DND [31] and SSID [1]. These datasets capture images with different noise levels, for instance by changing the ISO setting at capture. More related to our work, Zhang *et al.* [49] collect a dataset of microscopy images. All three datasets are designed only for denoising, and no HR images are provided that would allow them to be used for SR evaluation. According to our benchmark results, the best denoising algorithm does not necessarily provide the best input for the downstream SR task, and the JDSR learning is the best overall approach. This suggests a dataset on joint denoising and SR can provide a more comprehensive benchmark for image restoration.

**Real Super-resolution Dataset**  Recently, capturing LR and HR image pairs by changing camera parameters has been proposed. Chen *et al.* collect 100 pairs of images of printed postcards placed at different distances. SR-RAW [48] consists of 500 real scenes captured with multiple focal lengths. Although this dataset provides real LR-HR pairs, it suffers from misalignment due to the inevitable perspective changes or lens distortion. Cai *et al.* thus introduce an

iterative image registration scheme into the registration of another dataset, RealSR [4]. However, to have high-quality images, all these datasets are captured with low ISO setting, and the images thus contain very little noise as shown in our analysis. Qian *et al.* propose a dataset for joint demosaicing, denoising and SR [32], but the noise in their dataset is simulated by adding white Gaussian noise. Contrary to these datasets, our proposed W2S is constructed using SR microscopy techniques [15], all pairs of images are well aligned, and it contains raw LR images with different noise levels and the noise-free LR images, thus enabling the benchmarking of both denoising and SR under real settings.

### 2.3   Deep Learning based Image Restoration

Deep learning-based methods have shown promising results on various image restoration tasks, including denoising and SR. We briefly present prior work and the existing problems that motivate joint optimization.

**Deep Learning for Denoising** Recent deep learning approaches for image denoising achieve state-of-the-art results on recovering the noise-free images from images with additive noise, such as Gaussian noise. They outperform classical methods like BM3D [7]. Whether based on residual learning [45], using memory blocks [38], attention mechanisms [2], or internally modeling Gaussian noise parameters [11], these deep learning methods all require training data. For real-world raw-image denoising, the training data should include noisy images with a Poisson noise component, and a corresponding aligned noise-free image, which is not easy to acquire. However, these networks are evaluated only on the denoising task, often only on the one they are trained on. They optimize for minimal squared pixel error, leading to potentially smoothed out results that favour reconstruction error at the expense of detail preservation. When a subsequent task such as SR is then applied on the denoised outputs from these networks, the quality of the final results does not, as we see in our benchmark, necessarily correspond to the denoising performance of the different approaches. This highlights the need for a more comprehensive perspective that jointly considers both restoration tasks.

**Deep Learning for Super-resolution** Since the first convolutional neural network for SR [9] outperformed conventional methods on synthetic datasets, many new architectures [20,24,35,40,42,50,51] and loss functions [19,22,34,47,52] have been proposed to improve the effectiveness and the efficiency of the networks. To enable the SR networks generalize better on the real-world LR images where the degradation is unknown, works have been done on kernel prediction [4,14] and kernel modeling [46,54]. However, most of the SR networks assume that the LR images are noise-free or contain additive Gaussian noise with very small variance. Their predictions are easily affected by noise if the distribution of the noise is different from their assumptions [6]. This again motivates a joint approach developed for the denoising and SR tasks.

**Joint Optimization in Deep Image Restoration** Recent studies have shown the performance of joint optimization in image restoration, for example, the joint demosaicing and denoising [13,21], joint demosaicing and super-

resolution [48,53]. All these methods show that the joint solution outperforms the sequential application of the two stages. More relevant to JDSR, Xie *et al.* [43] present a dictionary learning approach with constraints tailored for depth maps, and Miao *et al.* [29] propose a cascade of two networks for joint denoising and deblurring, evaluated on synthetic data only. Similarly, our results show that a joint solution for denoising and SR also obtains better results than any sequential application. Note that our W2S dataset allows us to draw such conclusions on *real* data, rather than degraded data obtained through simulation.

## 3   Joint Denoising and Super-Resolution Dataset for Widefield to SIM Mapping

In this section, we describe the experimental setup that we use to acquire the sets of LR and HR images and present an analysis of the noise levels and blur kernels of our dataset.

### 3.1   Structured-Illumination Microscopy

Structured-illumination microscopy (SIM) is a technique used in microscopy imaging that allows samples to be captured with a higher resolution than the one imposed by the physical limits of the imaging system [15]. Its operation is based on the interference principle of the Moiré effect. We present how SIM works in more detail in our supplementary material. We use SIM to extend the resolution of standard widefield microscopy images. This allows us to obtain aligned LR and HR image pairs to create our dataset. The acquisition details are described in the next section.

### 3.2   Data Acquisition

We capture the LR images of the W2S dataset using widefield microscopy [41]. Images are acquired with a high-quality commercial fluorescence microscope and with real biological samples, namely, human cells.

**Widefield Images**  A time-lapse widefield of 400 images is acquired using a Nikon SIM setup (Eclipse T1) microscope. The details of the setup are given in the supplementary material. In total, we capture 120 different fields-of-view (FOVs), each FOV with 400 captures in 3 different wavelengths. All images are *raw*, *i.e.*, are linear with respect to focal plane illuminance, and are made up of $512 \times 512$ pixels. We generate different noise-level images by averaging 2, 4, 8, and 16 raw images of the same FOV. The larger the number of averaged raw images is, the lower the noise level. The noise-free LR image is estimated as the average of all 400 captures of a single FOV. Examples of images with different noise levels and the corresponding noise-free LR images are presented in Fig. 1.

**SIM Imaging**  The HR images are captured using SIM imaging. We acquire the SIM images using the same Nikon SIM setup (Eclipse T1) microscope as above. We present the details of the setup in the supplementary material. The HR

images have a resolution that is higher by a factor of 2, resulting in $1024 \times 1024$ pixel images.

### 3.3  Data Analysis

W2S includes 120 different FOVs, each FOV is captured in 3 channels, corresponding to the wavelengths 488nm, 561nm and 640nm. As the texture of the cells is different and independent across different channels, the different channels can be considered as different images, thus resulting in 360 views. For each view, 1 HR image and 400 LR images are captured. We obtain LR images with different noise levels by averaging different numbers of images of the same FOV and the same channel. In summary, W2S provides 360 different sets of images, each image set includes LR images with 5 different noise levels (corresponding to 1, 2, 4, 8, and 16 averaged LR images), the corresponding noise-free LR image (averaged over 400 LR images) and the corresponding HR image acquired with SIM. The LR images have dimensions $512 \times 512$, and the HR images $1024 \times 1024$.

To quantitatively evaluate the difficulty of recovering the HR image from the noisy LR observation in W2S, we analyze the degradation model relating the LR observations to their corresponding HR images. We adopt a commonly used degradation model [5,9,14,54], with an additional noise component,

$$I_{LR}^{noisy} = (I_{HR} \circledast k) \downarrow_m + n, \tag{1}$$

where $I_{LR}^{noisy}$ and $I_{HR}$ correspond, respectively, to the noisy LR observation and the HR image, $\circledast$ is the convolution operation, $k$ is a blur kernel, $\downarrow_m$ is a downsampling operation with a factor of $m$, and $n$ is the additive noise. Note that $n$ is usually assumed to be zero in most of the SR networks' degradation models, while it is not the case for our dataset. As the downsampling factor $m$ is equal to the targeted super-resolution factor, it is well defined for each dataset. We thus analyze in what follows the two unknown variables of the degradation model for W2S; namely the noise $n$ and the blur kernel $k$. Comparing to other denoising datasets, W2S contains 400 noisy images for each view, DND [6] contains only 1, SSID [1] contains 150, and FMD [49], which also uses widefield imaging, contains 50. W2S can thus provide a wide range of noise levels by averaging a varying number of images out of the 400. In addition, W2S provides LR and HR image pairs that do not suffer from misalignment problems often encountered in SR datasets.

**Noise Estimation**  We use the noise modeling method in [12] to estimate the noise magnitude in raw images taken from W2S, from the denoising dataset FMD [49], and from the SR datasets RealSR [4] and City100 [5]. The approach of [12] models the noise as Poisson-Gaussian. The measured noisy pixel intensity is given by $y = x + n_P(x) + n_G$, where $x$ is the noise-free pixel intensity, $n_G$ is zero-mean Gaussian noise, and $x + n_P(x)$ follows a Poisson distribution of mean $ax$ for some $a > 0$. This approach yields an estimate for the parameter $a$ of the Poisson distribution. We evaluate the Poisson parameter of the noisy images from the three noise levels (obtained by averaging 1, 4 and 8 images)

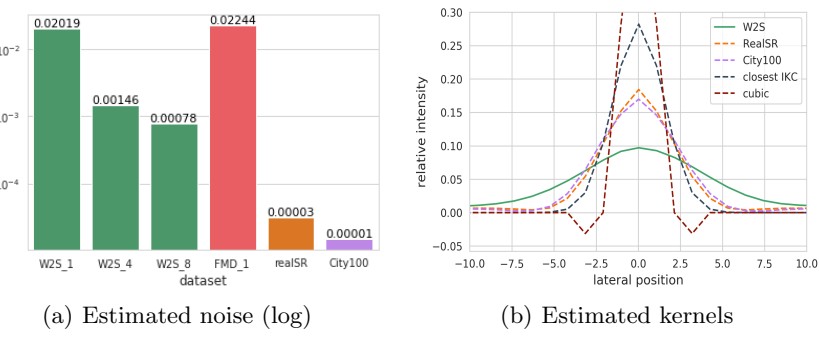

(a) Estimated noise (log)          (b) Estimated kernels

**Fig. 2.** Noise and kernel estimation on images from different datasets. A comparably-high noise level and a wide kernel indicate that the HR images of W2S are challenging to recover from the noisy LR observation.

of W2S, the raw noisy images of FMD, and the LR images of the SR datasets for comparison. We show the mean of the estimated noise magnitude for the different datasets in Fig. 2(a). We see that the raw noisy images of W2S have a high noise level, comparable to that of FMD. On the other hand, the estimated noise parameters of the SR datasets are almost zero, up to small imprecision, and are thus significantly lower than even the estimated noise magnitude of the LR images from the lowest noise level in W2S. Our evaluation highlights the fact that the additive noise component is not taken into consideration in current state-of-the-art SR datasets. The learning-based SR methods using these datasets are consequently not tailored to deal with noisy inputs that are common in many practical applications, leading to potentially poor performance. In contrast, W2S contains images with high (and low) noise magnitude comparable to the noise magnitude of a recent denoising dataset [49].

**Blur Kernel Estimation**  We estimate the blur kernel $k$ shown in Eq. (1) as

$$k = \underset{k}{argmin}||I_{LR}^{noise-free} \uparrow^{bic} -k \circledast I_{HR}||_2^2, \tag{2}$$

where $I_{LR}^{noise-free} \uparrow^{bic}$ is the noise-free LR image upscaled using bicubic inter-polation. We solve for $k$ directly in the frequency domain using the Fast Fourier Transform [10]. The estimated blur kernel is visualized in Fig. 2(b). For the pur-pose of comparison, we show the estimated blur kernel from two SR datasets: RealSR [4] and City100 [5]. We also visualize the two other blur kernels: the MATLAB bicubic kernel that is commonly used in the synthetic SR datasets, and the Gaussian blur kernel with a sigma of 2.0, which is the largest kernel used by the state-of-the-art blind SR network [14] for the upscaling factor of 2. From the visualization we clearly see the bicubic kernel and Gaussian blur kernel that are commonly used in synthetic datasets are very different from the blur kernels of real captures. The blur kernel of W2S has a long tail compared to the blur kernels estimated from the other SR datasets, illustrating that more

high-frequency information is removed for the LR images in W2S. This is because a wider space-domain filter corresponds to a narrower frequency-domain low pass, and vice versa. Hence, the recovery of HR images from such LR images is significantly more challenging.

Compared to the SR datasets, the LR and HR pairs in W2S are well-aligned during the capture process, and no further registration is needed. Furthermore, to obtain high-quality images, the SR datasets are captured under high ISO and contain almost zero noise, whereas W2S contains LR images with different noise levels. This makes it a more comprehensive benchmark for testing under different imaging conditions. Moreover, as shown in Sec. 3.3, the estimated blur kernel of W2S is wider than that of other datasets, and hence it averages pixels over a larger window, filtering out more frequency components and making W2S a more challenging dataset for SR.

## 4    Benchmark

We benchmark on the sequential application of state-of-the-art denoising and SR algorithms on W2S using RMSE and SSIM, which are two common metrics for evaluating image quality. Note that we do not consider the inverse order, *i.e.*, first applying SR methods on noisy images, as this amplifies the noise and causes a large increase in RMSE as shown in the last row of Table 2. With current methods, it would be extremely hard for a subsequent denoiser to recover the original clean signal.

### 4.1    Setup

We split W2S into two disjoint training and test sets. The training set consists of 240 LR and HR image sets, and the test set consists of 120 sets of images, with no overlap between the training set and the test set. We retrain the learning-based methods on the training set, and the evaluation of all methods is carried out on the test set.

For denoising, we evaluate different approaches from both classical methods and deep-learning methods. We use a method tailored to address Poisson denoising, PURE-LET [25], and the classical Gaussian denoising methods EPLL [55] and BM3D [7]. The Gaussian denoisers are combined with the Anscombe variance-stabilization transform (VST) [26] to first modify the distribution of the image noise into a Gaussian distribution, denoise, and then invert the result back with the inverse VST. We estimate the noise magnitude using the method in [12], to be used as input for both the denoiser and for the VST when the latter is needed. We also use the state-of-the-art deep-learning methods MemNet [38], DnCNN [45], and RIDNet [2]. For a fair comparison with the traditional non-blind methods that are given a noise estimate, we separately train each of these denoising methods for every noise level, and test with the appropriate model per noise level. The training details are presented in the supplementary material.

| | Method | \multicolumn{5}{c}{Number of raw images averaged before denoising} | | | | |
| :-: | :-- | :-: | :-: | :-: | :-: | :-: |
| | | 1 | 2 | 4 | 8 | 16 |
| Denoisers | PURE-LET [25] | 0.089/0.864 | 0.076/0.899 | 0.062/0.928 | 0.052/0.944 | 0.044/0.958 |
| | VST+EPLL [55] | 0.083/0.887 | 0.074/0.916 | 0.061/0.937 | 0.051/0.951 | 0.044/0.962 |
| | VST+BM3D [7] | 0.080/0.897 | 0.072/0.921 | 0.059/0.939 | 0.050/0.953 | 0.043/0.962 |
| | MemNet† [38] | 0.090/0.901 | 0.072/0.909 | 0.063/0.925 | 0.059/0.944 | 0.059/0.944 |
| | DnCNN† [45] | 0.078/0.907 | 0.061/0.926 | 0.049/0.944 | 0.041/0.954 | 0.033/0.964 |
| | RIDNet† [2] | 0.076/0.910 | 0.060/0.928 | 0.049/0.943 | 0.041/0.955 | 0.034/0.964 |

**Table 1.** RMSE/SSIM results on denoising the W2S test images. We benchmark a variety of standard methods, three classical ones (of which PURE-LET is designed for Poisson noise removal), and three deep learning based methods. The larger the number of averaged raw images is, the lower the noise level. †These learning-based methods are trained for each noise level separately, on our training set. An interesting observation is that the best RMSE results (in red) do not necessarily give the best result after the downstream SR method, as we see in Table 2. We highlight the results under the highest noise level with gray background for easier comparison with Table 2.

We use six state-of-the-art SR networks for the benchmark: four pixel-wise distortion based SR networks, RCAN [50], RDN [51], SAN [8], SRFBN [23], and two perceptually-optimized SR networks, EPSR [40] and ESRGAN [42]. The networks are trained for SR and the inputs are assumed to be noise-free, *i.e.*, they are trained to map from the noise-free LR images to the high-quality HR images. All these networks are trained using the same settings, the details of which are presented in the supplementary material.

### 4.2    Results and Discussion

We apply the denoising algorithms on the noisy LR images, and calculate the RMSE and SSIM values between the denoised image and the corresponding noise-free LR image in the test set of W2S. The results of the 6 benchmarked denoising algorithms are shown in Table 1. DnCNN and RIDNet outperform the classical denoising methods for all noise levels. Although MemNet achieves worse results than the classical denoising methods in terms of RMSE and SSIM, the results of MemNet contain fewer artifacts as shown in Fig. 3.

One interesting observation is that a better denoising with a lower RMSE or a higher SSIM, in some cases, results in unwanted smoothing in the form of a local filtering that incurs a loss of detail. Although the RMSE results of DnCNN are not the best (Table 1), when they are used downstream by the SR networks in Table 2, the DnCNN denoised images achieve the best final performance.

Qualitative denoising results are shown in the first row of Fig. 3. We note that the artifacts created by denoising algorithms are amplified when SR methods are applied on the denoised results (*e.g.*, (a) and (b) of Fig. 3). Although the denoised images are close to the clean LR image according to the evaluation metrics, the SR network is unable to recover faithful texture from these denoised images as the denoising algorithms remove part of the high-frequency information.

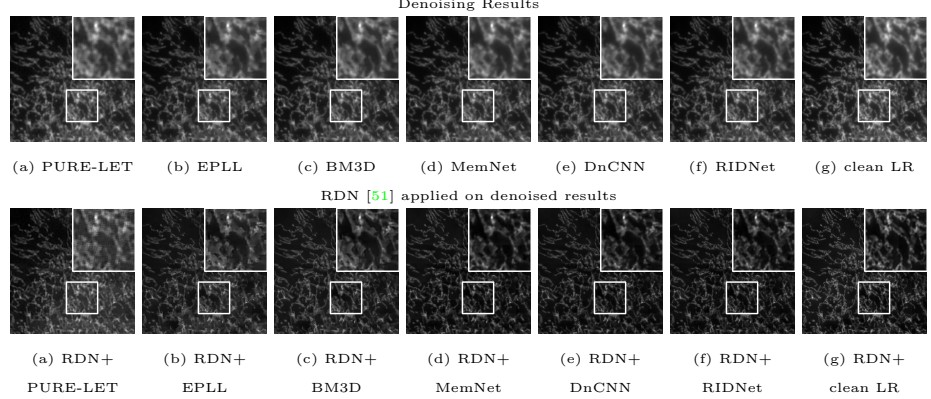

**Fig. 3.** The first row shows qualitative results of the denoising algorithms on a test LR image with the highest noise level. The second row shows qualitative results of the SR network RDN [51] applied on top of the denoised results. RDN amplifies the artifacts created by PURE-LET [25] and EPLL [55], and is unable to recover faithful texture when the input image is over-smoothed by denoising algorithms. A gamma correction is applied for better visualization. Best viewed on screen.

The SR networks are applied on the denoised results of the denoising algorithms, and are evaluated using RMSE and SSIM. We also include the results of applying the SR networks on the noise-free LR images. As mentioned above, we notice that there is a significant drop in performance when the SR networks are given the denoised LR images instead of the noise-free LR images as shown in Table 1. For example, applying RDN on noise-free LR images results in the SSIM value of 0.836, while the SSIM value of the same network applied to the denoised results of RIDNet on the lowest noise level is 0.756 (shown in the first row, last column in Table 3). This illustrates that the SR networks are strongly affected by noise or over-smoothing in the inputs. We also notice that a better SR network according to the evaluation on a single SR task does not necessarily provide better final results when applied on the denoised images. Although RDN outperforms RCAN in both RMSE and SSIM when applied on noise-free LR images, RCAN is more robust when the input is a denoised image. Among all the distortion-based SR networks, RCAN shows the most robustness as it outperforms all other networks in terms of RMSE and SSIM when applied on denoised LR images. As mentioned above, another interesting observation is that although DnCNN results in lower RMSE and higher SSIM than other networks for denoising at the highest noise level, DnCNN still provides a better input for the SR networks. We note generally that better denoisers according to the denoising benchmark do not necessarily provide better denoised images for the downstream SR task. Although the denoised results from MemNet have larger RMSE than the conventional methods, as shown in Table 1, the SR results on MemNet's denoised images achieve higher quality based on RMSE and SSIM.

|  | | Super-resolution networks | | | | | |
|---|---|---|---|---|---|---|---|
|  |  | RCAN | RDN | SAN | SRFBN | EPSR | ESRGAN |
| Denoisers | PURE-LET | .432/.697 | .458/.695 | .452/.693 | .444/.694 | .658/.594 | .508/.646 |
|  | VST+EPLL | .425/.716 | .434/.711 | .438/.707 | .442/.710 | .503/.682 | .485/.703 |
|  | VST+BM3D | .399/.753 | .398/.748 | .418/.745 | .387/.746 | .476/.698 | .405/.716 |
|  | MemNet | .374/.755 | .392/.749 | .387/.746 | .377/.752 | .411/.713 | .392/.719 |
|  | DnCNN | .357/.756 | .365/.749 | .363/.753 | .358/.754 | .402/.719 | .373/.726 |
|  | RIDNet | .358/.756 | .371/.747 | .364/.752 | .362/.753 | .411/.710 | .379/.725 |
|  | Noise-free LR | .255/.836 | .251/.837 | .258/.834 | .257/.833 | .302/.812 | .289/.813 |
|  | Noisy LR | .608/.382 | .589/.387 | .582/.388 | .587/.380 | .627/.318 | .815/.279 |

**Table 2.** RMSE/SSIM results on the sequential application of denoising and SR methods on the W2S test images with the highest noise level, corresponding to the first column of Table 1. We omit the leading '0' in the results for better readability. For each SR method, we highlight the best RMSE value in red. The SR networks applied on the denoised results are trained to map the noise-free LR images to the high-quality HR images. Results on other noise levels are presented in the supplementary material.

Qualitative results are given in Fig. 4, where for each SR network we show the results for the denoising algorithm that achieves the highest RMSE value for the joint task (*i.e.*, using the denoised results of DnCNN). We note that none of networks is able to produce results with detailed texture. As denoising algorithms remove some high-frequency signals along with noise, the SR results from the distortion-based networks are blurry and many texture details are lost. Although the perception-based methods (EPSR and ESRGAN) are able to produce sharp results, they fail to reproduce faithful texture and suffer a drop in SSIM.

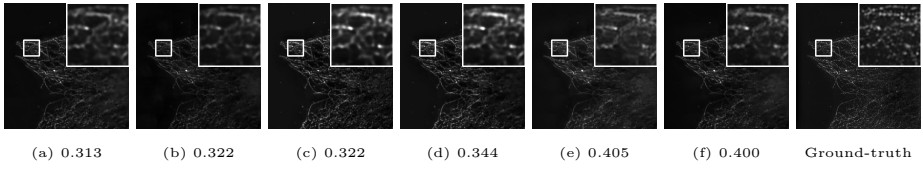

(a) 0.313      (b) 0.322      (c) 0.322      (d) 0.344      (e) 0.405      (f) 0.400      Ground-truth

**Fig. 4.** Qualitative results with the corresponding RMSE values on the sequential application of denoising and SR algorithms on the W2S test images with the highest noise level. (a) DnCNN [45]+RCAN [50], (b) DnCNN [45]+RDN [51], (c) DnCNN [45]+SAN [8], (d) DnCNN [45]+SRFBN [23], (e) DnCNN [45]+EPSR [40], (f) DnCNN [45]+ESRGAN [42]. A gamma correction is applied for better visualization. Best viewed on screen.

## 4.3   Joint Denoising and Super-Resolution (JDSR)

Our benchmark results in Sec. 4 show that the successive application of denoising and SR algorithms does not produce the highest-quality HR outputs. In this

section, we demonstrate that it is more effective to train a JDSR model that directly transforms the noisy LR image into an HR image.

### 4.4   Training Setup

For JDSR, we adopt a 16-layer RRDB network [42]. To enable the network to better recover texture, we replace the GAN loss in the training with a novel texture loss. The GAN loss often results in SR networks producing realistic but fake textures that are different from the ground-truth and may result in a significant drop in SSIM [42]. Instead, we introduce a texture loss that exploits the features' second-order statistics to help the network produce high-quality and real textures. This choice is motivated by the fact that traditional second-order descriptors have proven particularly effective for tasks such as texture recognition [16]. We leverage the difference in second-order statistics of VGG features to measure the similarity of the texture between the reconstructed HR image and the ground-truth HR image. Our texture loss is defined as

$$\mathcal{L}_{texture} = ||Cov(\phi(I_{SR})) - Cov(\phi(I_{HR}))||_2^2, \tag{3}$$

where $I_{SR}$ is the estimated result from the network for JDSR and $I_{HR}$ is the ground-truth HR image, $\phi(\cdot)$ is a neural network feature space, and $Cov(\cdot)$ computes the covariance. We follow the implementation of MPN-CONV [30] for the forward and backward feature covariance calculation. To improve visual quality, we further incorporate a perceptual loss to the training objective

$$\mathcal{L}_{perceptual} = ||\phi(I_{SR}) - \phi(I_{HR})||_2^2. \tag{4}$$

Our final loss function is then given by

$$\mathcal{L} = \mathcal{L}_1 + \alpha \cdot \mathcal{L}_{perceptual} + \beta \cdot \mathcal{L}_{texture}, \tag{5}$$

where $\mathcal{L}_1$ represents the $\ell 1$ loss between the estimated image and the ground-truth. We empirically set $\alpha = 0.05$ and $\beta = 0.05$. For the neural network feature space, we use a pre-trained 19-layer VGG [37].

We follow the same training setup as the experiments in Sec. 4. For comparison, we also train RCAN [51] and ESRGAN [42] on JDSR.

### 4.5   Results and Discussion

The quantitative results of different methods on different noise levels are reported in Table 3. The results indicate that comparing to the sequential application of denoising and SR, a single network trained on JDSR is more effective even though it has fewer parameters. GAN-based methods generate fake textures and lead to low PSNR and SSIM scores. Our model, trained with texture loss, is able to outperform RDN and effectively recover high-fidelity texture information even when high noise levels are present in the LR inputs. We show the qualitative results of JDSR on the highest noise level (which corresponds to the first column of Table 1) in Fig. 5. We see that other networks have difficulties to recover the shape of the cells in the presence of noise, whereas our method trained with texture loss is able to generate a higher-quality HR image with faithful texture.

| Method | Number of raw images averaged before JDSR | | | | #Parameters |
| | 1 | 2 | 4 | 8 | |
|---|---|---|---|---|---|
| DnCNN[†]+RCAN[‡] | 0.357/0.756 | 0.348/0.779 | 0.332/0.797 | 0.320/0.813 | 0.5M+15M |
| DnCNN[†]+ESRGAN[‡] | 0.373/0.726 | 0.364/0.770 | 0.349/0.787 | 0.340/0.797 | 0.5M+18M |
| JDSR-RCAN[*] | 0.353/0.767 | 0.340/0.780 | 0.324/0.799 | 0.318/0.814 | 15M |
| JDSR-ESRGAN[*] | 0.361/0.758 | 0.359/0.771 | 0.346/0.788 | 0.332/0.798 | 18M |
| Ours[*] | 0.357/0.760 | 0.346/0.779 | 0.328/0.797 | 0.330/0.801 | 11M |

**Table 3.** JDSR RMSE/SSIM results on the W2S test set. [†]The denoising networks are retrained per noise level. [‡]The SR networks are trained to map noise-free LR images to HR images. [*]The networks trained for JDSR are also retrained per noise level.

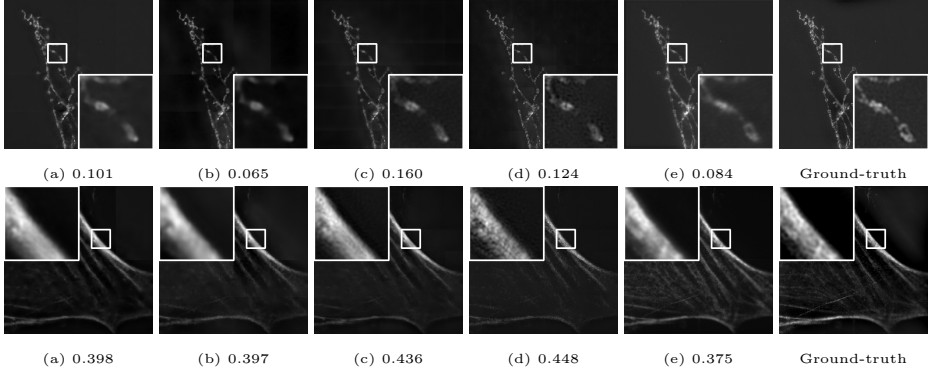

(a) 0.101      (b) 0.065      (c) 0.160      (d) 0.124      (e) 0.084      Ground-truth

(a) 0.398      (b) 0.397      (c) 0.436      (d) 0.448      (e) 0.375      Ground-truth

**Fig. 5.** Qualitative results with the corresponding RMSE values of denoising and SR on the W2S test images with the highest noise level. (a) DnCNN+RCAN, (b) RCAN, (c) DnCNN+ESRGAN, (d) ESRGAN, (e) a 16-layer RRDB network [42] trained with texture loss. The multi-channel images are formed by mapping the three single-channel images of different wavelengths to RGB. A gamma correction is applied for better visualization. Best viewed on screen.

## 5    Conclusion

We propose the first joint denoising and SR microscopy dataset, **W**idefield**2S**IM. We use image averaging to obtain LR images with different noise levels and the noise-free LR. The HR images are obtained with SIM imaging. With W2S, we benchmark the combination of various denoising and SR methods. Our results indicate that SR networks are very sensitive to noise, and that the consecutive application of two approaches is sub-optimal and suffers from the accumulation of errors from both stages. We also observe form the experimental results that the networks benefit from joint optimization for denoising and SR. W2S is publicly available, and we believe it will be useful in advancing image restoration in medical imaging. Although the data is limited to the domain of microscopy data, it can be a useful dataset for benchmarking deep denoising and SR algorithms.

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
