# W2S: Microscopy Data with Joint Denoising and Super-Resolution for Widefield to SIM Mapping Supplementary Material

## 1 Structured-Illumination Microscopy (SIM)

We provide an introduction to the principle of structured interference acquisition with 1D signals. The extension to higher dimensions follows the same principle [1].

We define $s(t)$ to be the signal we want to acquire, where $t$ represents a certain spatial dimension. In the Fourier domain, the corresponding signal $S(\omega)$ is not necessarily band-limited and can be non-zero for arbitrary frequencies $\omega$. The impulse response of the capturing system is called its point spread function, and its Fourier transform is its optical transfer function (OTF) that we call $F(\omega)$. The resulting visible signal through that imaging system is given by $V(\omega) = F(\omega) \odot S(\omega)$, where $\odot$ is the element-wise multiplication. The OTF limits the captured content to a certain range of frequency components as $F(\omega) = 0 \ \forall \omega > \omega_c$, where $\omega_c$ is the cut-off frequency of the OTF. Therefore, only frequency components $\omega < \omega_c$ can be captured. By using a structured-illumination pattern, the frequency content can be manipulated. For instance, if we multiply the signal $s(t)$ with a cosine function of frequency $\omega_0$, the captured signal becomes $s(t) \odot cos(\omega_0 t)$ and the corresponding frequency-domain equivalent is given by $\frac{1}{2}[S(\omega - \omega_0) + S(\omega + \omega_0)]$. Using the same imaging system, the visible signal becomes

$$V(\omega) = F(\omega) \odot \frac{1}{2}[S(\omega - \omega_0) + S(\omega + \omega_0)], \tag{1}$$

with $F(\omega)$ still equal to zero above its cut-off frequency $\omega_c$. However, frequency components such that $\omega - \omega_0 < \omega_c$ can now be acquired, effectively pushing the cut-off to $\omega_c + \omega_0$, where $\omega_0$ can be controlled by modifying the periodicity of the illumination pattern. In other words, higher frequencies that could not be visible to the imaging system can be shifted down to a lower range that lies within the observable range of that system. The shifted components can overlap in the frequency domain, and multiple shifted captures are needed to resolve the ambiguity and recover the true signals. SIM acquires 15 different structured-illumination images to perform an upscaling by a factor of two. In practice, the OTF is also not necessarily an ideal low-pass filter and a deconvolution post-processing step can be required.

Note that theoretically, applying nonlinear SIM can produce images of arbitrary resolution [2], illustrating SIM's potential of producing a higher upscaling-factor SR dataset.

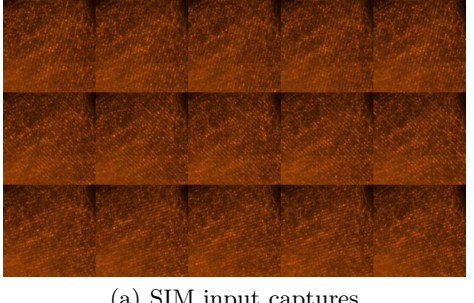
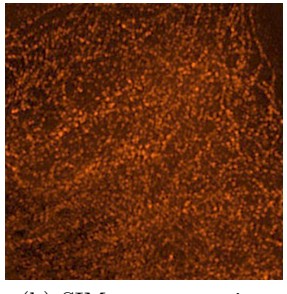

(a) SIM input captures          (b) SIM reconstruction

**Fig. 1.** Example FOV showing the different images captured under structured illumination in (a), which are given as input to the SIM method, and the reconstructed result of SIM in (b). Gamma correction is applied for better visualization.

## 2 Acquisition Details

**Widefield Images** To capture the widefield images, the microscope is fitted with a Plan Apochromat TIRF 100X, 1.49NA objective and an electron-multiplying charge-coupled device camera (IXON3; Andor Technology). The acquisition is taken with a 5ms exposure time using a 488nm Coherent sapphire laser at 0.37mW, a 5ms exposure time using a 561nm Cobolt Laser at 0.28mW, and a 5ms exposure using a Coherent 640nm Cobolt Laser at 0.26mW. In total, we capture 120 different fields-of-view (FOVs) on 3 different wavelength values, and each FOV is repeatedly captured 400 consecutive times. All images are $512 \times 512$ pixels.

**SIM Imaging** The SIM images are captured using the same device (a microscope fitted with a Plan Apochromat TIRF 100X). We use the 3D SIM acquisition mode [3] (15 images per plane; five phases of three rotations) with a 70ms exposure time, using a 488nm Coherent sapphire laser at 0.20mW; 30ms exposure time using a 561nm Cobolt Laser at 0.27mW, and a 100ms exposure using a Coherent 640nm Cobolt Laser at 0.14mW. Image reconstruction and processing are performed using the NIS-Elements software. An example of the acquired images and the reconstructed HR result is shown in Fig. 1.

## 3 Training Details

**Data normalization** To facilitate the training of the networks, we apply z-score normalization on the LR images. For each raw image captured by the widefield microscope, the normalized image is given by

$$I_{LR}^{normalized} = \frac{I_{LR} - \mu}{\sigma}, \tag{2}$$

where $\mu$ and $\sigma$ are, respectively, the mean value and the standard deviation across all LR raw images. To match the intensity of the SIM images to the

intensity of the corresponding LR images, we apply a scaling and shift on the image pixels

$$I_{HR}^{normalized} = a * I_{HR} + b, \tag{3}$$

where

$$a, b = \underset{a,b}{\text{argmin}} ||[a * (I_{HR} \downarrow 2) + b] - I_{LR}^{normalized}||_2^2, \tag{4}$$

and Eqn. 4 is solved using least square regression.

**Denoisers** DnCNN and MemNet use a batch size of 128 and a starting learning rate of $10^{-3}$, while RIDNet uses batches of 64 patches and a starting learning rate of $5 \times 10^{-4}$, all trained with the Adam optimizer [4] for 50 epochs, and with a ten-fold decrease in the learning rate after the milestone of 30 epochs. The same settings are used when training for the noise levels corresponding to an average of 1, 2, 4, 8, and 16 normalized raw images.

**Super-resolution networks** For the super-resolution (SR) networks we benchmark, we train with the initial learning rate and loss function described in the corresponding papers. For fair comparison, we use the same training setup for all models. For each training batch, 16 LR patches of size $64 \times 64$ are extracted. All models are trained using the Adam optimizer [4] for 50 epochs. The learning rate decreases by half every 10 epochs. Data augmentation is performed on the training images with a probability of 0.5. They are randomly rotated by $90°$, flipped horizontally, or flipped vertically.

## 4    Additional Benchmark Results

We present additional benchmark results of the sequential application of state-of-the-art denoisers and SR methods on low-resolution (LR) images with different noise levels on W2S. The different noise levels correspond to a different number of averaged raw images.

We note that there is no consistent and significantly-better denoiser across all SR methods and noise levels. Between SR models, the best is RDN but not by a large margin. We only observe a consistent and significant improvement when we train networks end-to-end on our JDSR data, and further when we leverage the novel loss presented in the main paper (Table 3 of the paper).

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

| | | Super-resolution networks | | | | | |
|---|---|---|---|---|---|---|---|
| | | RCAN | RDN | SAN | SRFBN | EPSR | ESRGAN |
| Denoisers on 2× average | PURE-LET | 0.422/0.81 | 0.385/0.75 | 0.382/0.75 | 0.383/0.75 | 0.405/0.74 | 0.389/0.75 |
| | VST+EPLL | 0.381/0.79 | 0.374/0.77 | 0.367/0.76 | 0.375/0.77 | 0.390/0.75 | 0.381/0.76 |
| | VST+BM3D | 0.362/0.78 | 0.355/0.78 | 0.363/0.78 | 0.357/0.77 | 0.380/0.76 | 0.372/0.77 |
| | DnCNN† | 0.352/0.78 | 0.352/0.78 | 0.354/0.78 | 0.355/0.78 | 0.376/0.76 | 0.367/0.77 |
| | MemNet† | 0.348/0.78 | 0.349/0.78 | 0.344/0.78 | 0.349/0.78 | 0.369/0.76 | 0.364/0.77 |
| | RIDNet† | 0.348/0.78 | 0.346/0.78 | 0.347/0.78 | 0.351/0.78 | 0.374/0.76 | 0.364/0.77 |
| Denoisers on 4× average | PURE-LET | 0.406/0.83 | 0.369/0.77 | 0.367/0.77 | 0.369/0.77 | 0.391/0.75 | 0.374/0.77 |
| | VST+EPLL | 0.366/0.81 | 0.354/0.78 | 0.352/0.79 | 0.352/0.79 | 0.376/0.77 | 0.366/0.78 |
| | VST+BM3D | 0.346/0.80 | 0.344/0.79 | 0.345/0.79 | 0.340/0.80 | 0.363/0.78 | 0.357/0.78 |
| | DnCNN† | 0.336/0.80 | 0.334/0.79 | 0.336/0.79 | 0.336/0.80 | 0.354/0.78 | 0.352/0.79 |
| | MemNet† | 0.332/0.80 | 0.334/0.80 | 0.333/0.80 | 0.333/0.79 | 0.351/0.78 | 0.349/0.79 |
| | RIDNet† | 0.332/0.80 | 0.330/0.80 | 0.332/0.80 | 0.330/0.80 | 0.350/0.79 | 0.349/0.79 |
| Denoisers on 8× average | PURE-LET | 0.394/0.84 | 0.357/0.78 | 0.356/0.78 | 0.358/0.78 | 0.379/0.76 | 0.365/0.78 |
| | VST+EPLL | 0.354/0.83 | 0.343/0.80 | 0.343/0.80 | 0.342/0.80 | 0.365/0.78 | 0.357/0.79 |
| | VST+BM3D | 0.334/0.82 | 0.331/0.81 | 0.330/0.81 | 0.330/0.81 | 0.352/0.79 | 0.348/0.79 |
| | DnCNN† | 0.324/0.81 | 0.323/0.81 | 0.322/0.81 | 0.324/0.81 | 0.346/0.79 | 0.343/0.80 |
| | MemNet† | 0.320/0.81 | 0.319/0.81 | 0.321/0.81 | 0.321/0.81 | 0.342/0.80 | 0.340/0.80 |
| | RIDNet† | 0.320/0.81 | 0.320/0.81 | 0.321/0.81 | 0.318/0.81 | 0.342/0.80 | 0.340/0.80 |
| Denoisers on 16× average | PURE-LET | 0.380/0.84 | 0.342/0.79 | 0.342/0.79 | 0.343/0.79 | 0.360/0.77 | 0.347/0.78 |
| | VST+EPLL | 0.340/0.83 | 0.328/0.80 | 0.328/0.80 | 0.328/0.80 | 0.345/0.78 | 0.339/0.79 |
| | VST+BM3D | 0.320/0.82 | 0.315/0.81 | 0.316/0.81 | 0.315/0.81 | 0.333/0.79 | 0.330/0.80 |
| | DnCNN† | 0.310/0.82 | 0.308/0.82 | 0.310/0.81 | 0.309/0.81 | 0.327/0.80 | 0.325/0.80 |
| | MemNet† | 0.306/0.81 | 0.305/0.82 | 0.305/0.82 | 0.306/0.82 | 0.322/0.80 | 0.322/0.80 |
| | RIDNet† | 0.306/0.81 | 0.305/0.82 | 0.305/0.81 | 0.306/0.81 | 0.323/0.80 | 0.322/0.80 |

**Table 1.** PSNR $(dB)$/SSIM results on the sequential application of denoising and SR methods on the W2S test images for different noise levels. †The learning-based denoising methods are retrained for each noise level, and the SR networks are trained to map the noise-free LR images to the high-quality HR images.