# OpenReview forum: "W2S: Microscopy Data with Joint Denoising and Super-Resolution for Widefield to SIM Mapping"
_thecvf.com/ECCV/2020/Workshop/BIC — BIC 2020 Oral_

### Official Review · AnonReviewer1 · 2020-07-30
**An extensive new dataset and a new state-of-the-art algorithm for denoising and super-resolution.**

**Rating:** 9
**Confidence:** 3

**Review:**

The authors address the problem of denoising and image resolution improvement in microscopy. A new public benchmark dataset is presented, much more extensive than the currently available alternatives.The authors perform a detailed evaluation of the existing denoising and super-resolution algorithms and finally propose their own which handles both tasks and out-performs the others. All code and data is available.

**Reviews Visibility:**

I agree that my anonymized review is made publicly visible, if the submission is accepted.

---

### Official Review · AnonReviewer3 · 2020-07-31
**Potentially VERY useful public dataset, some questions regarding neglected related work**

**Rating:** 7
**Confidence:** 5

**Review:**

While the submitted manuscript is indeed aiming a a joint denoising and superres task, there is quite a body of work that was not included in the related work sections or in the comparative parts of the paper (CARE, N2V, PN2V, etc.).

I have the feeling that the utility of the data should be valued higher than the incompleteness of comparisons, but if the camera-ready version could at least acknowledge the broader existing literature it would certainly be a positive.

With respect to the data, it appears that the availability of it is still pending to some degree. From the GitHub repo of the paper:
```
To those who have cloned or forked our repository, we now removed the png data and are working with the raw data pre-processed only with a single global z-score normalization. All the consequent modifications are being made. The full raw data will be made public very soon, and pretrained models (with raw data) will be made available by mid July.
```

In my opinion it would be highly desirable to bring the work on this public dataset to completion together with the submission of the camera-ready version.


**Reviews Visibility:**

I agree that my anonymized review is made publicly visible, if the submission is accepted.

---

### Decision · Program_Chairs · 2020-07-31

Accept (Oral)